# Effects of High Hydrostatic Pressure and Storage Temperature on Fatty Acids and Non-Volatile Taste Active Compounds in Red Claw Crayfish (*Cherax quadricarinatus*)

**DOI:** 10.3390/molecules27165098

**Published:** 2022-08-11

**Authors:** Chunsheng Liu, Meng Li, Yuanyuan Wang, Yi Yang, Aimin Wang, Zhifeng Gu

**Affiliations:** 1State Key Laboratory of Marine Resources Utilization in South China Sea, Hainan University, Haikou 570228, China; 2Ocean College, Hainan University, Haikou 570228, China; 3Coconut Research Institute of Chinese Academy of Tropical Agricultural Sciences, Wenchang 570311, China

**Keywords:** crayfish, high hydrostatic pressure (HHP), cold storage, fatty acids, non-volatile taste active compounds

## Abstract

The effects of high hydrostatic pressure (treated with 200, 400 and 600 MPa) and storage temperatures (4 °C and −20 °C) on the fatty acids and flavor compounds of red claw crayfish were studied. HHP decreased the PUFA, GMP, IMP and AMP, citric and lactic acids, and PO_4_^3−^ contents, but the FAA, Ca^2+^ and Cl^−^ contents increased in HHP-treated crayfish compared to untreated crayfish at 0 d. Storage at −20 °C could restrain the fatty acids and flavor contents compared to those stored at 4 °C. The GMP, AMP, citric acid and PO_4_^3−^ contents decreased, and Ca^2+^ and Cl^−^ contents increased after storage at 4 °C for 15 d (*p* < 0.05). HHP at 200 and 400 MPa increased EUC on 0 d. No significant changes in EUC were observed after storage at −20 °C for 15 d, significant decreases were noted at 4 °C than the crayfish stored for 0 d (*p* < 0.05), except for the untreated group. Generally, HHP at 200 or 400 MPa, and storage at −20 °C is beneficial according to the shelling rates and EUC of crayfish.

## 1. Introduction

Crayfish, representing ~4% of world aquaculture by its market value, is one of the most important and popular aquatic food products worldwide because it is considered a delicacy and abundant nutrients [1]. Red claw crayfish (*Cherax quadricarinatus*) is one of the largest freshwater crayfish species (commercial size of 100–200 g) and is naturally distributed in the tropical regions of Southeast Papua New Guinea and Queensland in Australia [2,3]. This species has been cultured worldwide and has become an important freshwater aquatic product in aquaculture [4,5]. Since its introduction into China, the annual production of red claw crayfish has reached 3000 tons per year [6,7].

Recently, the consumption of shelled fresh crayfish has increased annually in China [8]. However, aquatic products can be easily contaminated with numerous human pathogens and spoilage organisms [8,9]. These deleterious microorganisms are considered the crucial influencing factors for the safety and quality of aquatic food and can raise health risks [10]. To eliminate these microorganisms, many efficient and internationally accepted post-harvesting processes have been developed, such as high hydrostatic pressure (HHP), ultrasound, irradiation, rapid chilling and chemical preservatives [11,12].

HHP is a non-thermal technology with the potential for food preservation and shelf-life extension while maintaining the natural characteristics of food [13,14]. For instance, the total viable counts in razor clams (*Sinonovacula constricta*) were reduced from 6.71 log_10_ colony-forming unit (CFU)/g to 1.54 log_10_ CFU/g at 400 MPa HHP treatment for 10 min [15]. The total plate counts in Pacific white shrimp (*Litopenaeus vannamei*) with HHP-treated (550 MPa/5 min/25 °C) were below the level of detection (undetected level, <1 log_10_ CFU/g) after storing at 4 °C and 25 °C for 15 d [16]. *Vibrio* spp. counts in Suminoe oyster (*Crassostrea ariakensis*) were significantly reduced from ~1 × 10^5^ CFU/g to undetectable levels at 400 MPa HHP treatment for 3 min [9]. Although the quality of aquatic food treated by HHP is less affected compared to conventional thermal processing, HHP, especially at higher pressure, also changes the appearance, nutrient and flavor compounds [11,17]. Recently, the effects of HHP on the quality of many aquatic animals, such as shrimps, crabs, oysters, bay scallops, squids, and fish, have been reported [18,19,20,21,22]. According to these studies, the optimum HHP conditions were various among different aquatic species. Therefore, it is vital to define proper HHP treatment conditions to improve food safety and prevent damage to the nutrition and flavor components in raw aquatic products.

Red claw crayfish has become one of the most important aquatic products due to its large size, high nutritional value and good taste [23]. Our previous study showed that HHP at ≥200 MPa could completely shuck the shell of red claw crayfish, and the counts of pathogenic *Vibrio* significantly decreased to an undetected level at ≥400 MPa [24]. However, few studies have reported the biochemical characteristics of this freshwater crayfish species [25,26]. Furthermore, no study has reported the changes in fatty acids and flavor compounds in red claw crayfish treated with different pressure levels and storage temperatures. Therefore, this study aimed to compare the effects of different pressure levels on the fatty acids and flavor compounds [free amino acid (FAA), 5′-nucleotides, organic acids and inorganic ions] in red claw crayfish and evaluate the stability of these characteristics in HHP-treated crayfish after storage at 4 °C and −20 °C for 15 d.

## 2. Results and Discussion

### 2.1. Fatty Acid Compositions

The fatty acid compositions of crayfish treated with different HHP, and storage temperatures are shown in Table 1. C20:5n-3 (EPA) was the major fatty acid, accounting for 18.55–23.49% of total fatty acid, followed by C18:1n-9 (16.59–19.62%) and C18:2n-6 (13.18–16.71%) in all treatment groups. Li et al. [26] reported that the top three fatty acids were EPA (19.72–24.36), C18:1n-9 (17.78–23.78%), and C16:0 (13.34–15.79%) in the muscle of female pre-adult red claw crayfish, while the top three fatty acids reported by Méndez-Martínez et al. [23] were C18:1n-9 (29.30–41.50%), C16:0 (19.54–27.35%) and C18:2n-6 (11.90–19.90%). Polyunsaturated fatty acid (PUFA) contents in red claw crayfish also showed obvious differences. A large fluctuation of fatty acid composition often occurs in aquatic animals, which can be affected by life stage, reproductive periodicity, and culture conditions [27]. In the present study, the body weight of crayfish was ~40 g, which showed significant differences in the other two reports (average body weights were 10.8 g and 19.25 g, respectively), causing the differences in fatty acid profiles.

As shown in Table 1, the EPA contents decreased after HHP treatment, and significant changes were observed when comparing 600 MPa-treated crayfish with the control group at 0 d (*p* < 0.05). On the contrary, the C18:1n-9 and C16:0 contents in crayfish increased after HHP treatment, and significant differences were observed when comparing 400- and 600-MPa treated crayfish with control groups (*p* < 0.05). The levels of PUFA decreased with the increase of pressure, while the monounsaturated fatty acids (MUFA) and total saturated fatty acids (SFA) showed opposite results. Significant differences were found between the untreated and 600 MPa-treated crayfish at 0 d (*p* < 0.05). Furthermore, the n-3 PUFAs (mainly contributed by EPA) in crayfish treated with 600 MPa showed a significant decrease compared to the untreated group (*p* < 0.05). In this study, the higher pressure (≥600 MPa) could cause some changes in the fatty acid profiles. Yi et al. [16] reported that the fat content of Pacific white shrimp also significantly decreased after HHP treatment (550 MPa/5 min), while there were only minor changes found in the compositions of fatty acid in oysters after HHP treatment [9,14]. Oysters have a stronger shell and can tolerate higher pressure compared to shrimps. Crayfish showed a cooked appearance after 600 MPa treatment [24], while only slight changes were observed in soft tissues of oysters treated with the same pressure [9]. The cooked phenomenon in 600 MPa-treated crayfish might cause the changing of fatty acid profiles.

Physical and biochemical changes occur in aquatic animals during chilled storage. In our study, the fatty acid compositions in all treatment groups showed almost no significant changes after 15-d storage at both 4 °C and −20 °C, except for some kinds of fatty acids in control and 200 MPa-treated crayfish at 4 °C. In control and 200 MPa-treated crayfish, the MUFA significantly increased from 21.49% and 21.50% to 23.44% and 23.29%, respectively, while the PUFA significantly decreased from 61.44% and 61.19% to 57.82% and 58.04%, respectively (*p* < 0.05). Similarly, decreases in PUFA composition were observed in Bogue (*Boops boops*) and raw oyster after ~two weeks of storage at 4 °C [9,28]. In our previous study, there were numerous spoilage organisms observed in untreated and 200 MPa-treated crayfish [24]. If these spoilage bacteria were not completely killed, the lipid could be deteriorated, which led to changes in fatty acid profiles [29].

### 2.2. Changes of FAAs and 5′-Nucleotide Contents in Crayfish

As shown in Table 2, twenty FAAs were identified in crayfish. Of these, arginine (Arg, 5.70–8.47 mg/g) content was the most abundant in all crayfish groups at 0 d, followed by glycine (Gly, 1.03–2.22 mg/g), glutamine (Gln, 0.99–1.76 mg/g), and alanine (Ala, 0.99–1.73 mg/g). According to the taste thresholds of different FAAs, the taste activity values (TAV) of Arg, Gly, Ala, and histidine (His) were more than 1 and considered active FAAs in crayfish [9]. As shown in previous reports, the main FAA contents were different in different aquatic species. For example, the top three FAAs were Arg, proline (Pro), and Ala in Japanese flying squid (*Todarodes pacificus*) muscles [21]; Gly, Ala, and glutamic acid (Glu) in Suminoe oysters [9]; Pro, Arg, and Gly in Chinese shrimp (*Fenneropenaeus chinensis*) and mud crab (*Scylla paramamosain*) [30,31]. Therefore, the differences in FAAs content contribute to various tastes of aquatic produces.

In comparing the effects of HHP on the FAA contents of crayfish muscle, we found that the total contents of umami amino acid (UAA) and bitter amino acid (BAA) in crayfish significantly increased after HHP treatment compared to the untreated group at 0 d (*p* < 0.05). FAAs are produced by proteolysis and certain amino acid metabolic pathways [21,32]. HHP can increase the contents of FAA according to protein denaturation, which has been reported in many aquatic species, such as the Suminoe oyster, squid, and cod (*Gadus morhua*) [9,21,33]. Furthermore, there were almost no significant differences in FAA contents of crayfish after storage for 15 d at 4 °C and −20 °C when compared to the crayfish treated with the same HHP levels, except for some kinds of FAA stored at 4 °C (significantly increased and decreased in tryptophan (Trp) and Arg, respectively, *p* < 0.05).

In previous reports, the tends of FFA contents were different after a period of storage. For example, the total FAA contents in squids significantly increased after 10-d storage at 4 °C (*p* < 0.05), while significant decreases in total FAA contents of Suminoe oyster were observed after 15-d storage at 4 °C and −20 °C (*p* < 0.05) [9,21]. In hammour (*Epinephelus coioides*), there was a slight decrease in total FAA contents of the HHP-treated group after 30-d storage at 4 °C when compared to those at 0 d (*p* > 0.05), which showed the same result as our study [34]. Generally, FAAs in an organism are produced from both proteolysis and certain amino acid metabolic pathways, in which proteolysis increases FAA contents while amino acid metabolism decreases the concentration of specific amino acids [32]. HHP treatment and refrigerated/frozen storage could modify this progress. Therefore, these discrepant contents of FFAs could be explained by the differences in the food system, storage temperature, pressure and substrates supplied by HHP-promoted proteolysis and metabolic rates [21].

As shown in Table 3, GMP, IMP, and AMP were the main 5′-nucleotide components in untreated crayfish (57.15, 42.27, and 51.80 mg/100 g, respectively). Furthermore, the contents of GMP, IMP, and AMP decreased after HHP treatment (except for IMP in the 200 MPa-treated group). Significant differences were observed in GMP content of 200, 400, and 600 MPa-treated groups, IMP contents of 400 and 600 MPa-treated groups, and AMP content in 600 MPa-treated group, compared to untreated crayfish at 0 d (*p* < 0.05). After 15-d storage at both 4 °C and −20 °C, the IMP contents in HHP-treated crayfish groups were almost the same as those at 0 d, while significant increases were observed in HHP-untreated crayfish (*p* < 0.05). However, there were no obvious differences in GMP and AMP contents of crayfish after storage for 15 d at −20 °C, compared to the same treated crayfish at 0 d. Significant decreases were observed in these two 5′-nucleotides when crayfish were stored at 4 °C for 15 d (*p* < 0.05).

GMP, IMP, and AMP are the three flavor-contributing 5′-nucleotides [9,14]. GMP provides a meaty flavor and can be used as a flavor enhancer [35]. IMP is an umami substance, and its taste can be strongly enhanced by some kinds of sweet amino acids, such as serine (Ser), Gly, and Ala [9]. AMP promotes umami and sweet taste in some seafood [21]. The decreases in 5′-nucleotides after HHP treatment and 4 °C storage were also observed in oysters, which showed the same tendency as our study [9,14].

### 2.3. Effect of HHP on EUC of Crayfish

As shown in Figure 1, the EUC values of crayfish treated with different HHP pressures and storage temperatures were calculated. The EUC values of crayfish first increased and then decreased with the enhancement of HHP pressure at 0 d. The highest EUC value was observed in 200 MPa-treated crayfish (3.92 g MSG/100 g wet weight), followed by 400 MPa-treated, untreated, and 600 MPa-treated individuals (3.11, 2.44, and 1.53 g MSG/100 g wet weight, respectively). Significant differences were observed among the four treatment crayfish groups (*p* < 0.05). Comparing the EUC values of crayfish with other aquatic produces, the values in untreated crayfish were similar to Chinese mitten crab (*Eriocheir sinensis*) (2.07–5.42 g MSG/100 g) [36], lower than raw oysters (6.47 g MSG/100 g) [9], and higher than Yangtze (*Coilia ectenes*) (0.33–0.99 g MSG/100 g) [37]. Furthermore, there were no significant differences in EUC values in crayfish stored for 15 d at −20 °C compared to the same treatment groups at 0 d. In comparison, significant decreases were found in crayfish at 4 °C for 15 d compared to those at 0 d (*p* < 0.05), except of the 600 MPa treatment group.

The EUC values were calculated by mixing two umami amino acids (Asp and Glu) and three 5′-nucleotides (GMP, IMP, and AMP). This parameter has been widely used to evaluate the umami taste of aquatic foods, such as crab, squid, and oyster [9,14,21,36]. In this study, Glu, IMP and GMP were the main umami compositions in crayfish, which influenced the EUC values. In addition, storage at −20 °C could significantly slow down the EUC values compared to 4 °C conditions for crayfish.

### 2.4. Comparison of Organic Acids and Betaine in Crayfish

As shown in Table 4, the concentrations of betaine were almost undetectable in untreated crayfish, while citric acid (59.82 mg/g), succinic acid (9.62 mg/g), and lactic acid (2.67 mg/g) were the predominant organic acids. The contents of citric and lactic acids significantly decreased after HHP pressure compared to untreated crayfish (*p* < 0.05), except for lactic acid in the 200 MPa-treated group. While the succinic acid slightly decreased after HHP pressure, significant differences were observed only in 600 MPa-treated crayfish compared to untreated ones (*p* < 0.05). In aquatic foods, the organic acids are mainly citric, lactic, acetic, malic, succinic, and propionic acids. The composition and concentrations of these organic acids are significantly different according to species and living conditions [30,38]. The major organic acids in Suminoe oyster were citric and succinic acids [9]; succinic and malic acids in *Catabacter hongkongensis* [14]; and succinic and lactic acids in mud crab [30]. Beside acidic taste, organic acids contribute to other flavors. For example, succinic acid could enhance bitter taste and strong salty at different concentrations [39]. Citric acid could contribute to oysters’ soft, crisp acidic taste [9].

The citric acid contents in all groups significantly decreased after storing for 15 d at 4 °C (*p* < 0.05). On the contrary, the malic acid content in all treatments and lactic acid content in control and 200 MPa treatment crayfish significantly increased (*p* < 0.05). For crayfish stored at −20 °C, significant decreases in lactic acid contents were observed in the control and 200 MPa treatment groups (*p* < 0.05). Furthermore, the citric acid contents in untreated crayfish significantly decreased to 36.75 mg/g of wet weight but still showed significantly higher than those at 4 °C (*p* < 0.05). Citric acid is the main substrate in the Krebs cycle of living organisms and undergoes degradation [40]. Therefore, its easy degradability caused by microorganisms may be the main reason for decreased citric acid in crayfish stored at 4 °C.

### 2.5. Changes of Inorganic Ions in Crayfish

Table 5 shows the contents of inorganic ions in crayfish. Among eight detected inorganic ions, the contents of PO_4_^3−^ were the highest, followed by K^+^, Cl^−^, Na^+^, Ca^2+,^ and Mg^2+^, and the trace element Zn^2+^. In previous studies, the contents of inorganic ions varied according to different aquatic species. In Yangtze, the PO_4_^3−^ content was the highest, followed by K^+^, Cl^−^, Na^+^, Mg^2+^, and Ca^2+^, which was similarly observed in crayfish in our study [38]. In Suminoe oyster, the contents of PO_4_^3−^, Cl^−^, Ca^2+^, Mg^2+^, Na^+^ and K^+^, showed a decreasing tendency [9]. In squids, the K^+^ content was the highest, followed by Na^+^, PO_4_^3−^, Mg^2+^, Ca^2+^, and Cl^−^ [21]. When comparing the content of Zn^2+^, Guo et al. [41] reported that these trace elements were 43.0 mg/kg in Chinese mitten crab, which was higher than those in crayfish.

The PO_4_^3−^ contents in HHP-treated crayfish were significantly lower than those in the untreated group (*p* < 0.05), while the opposite results were observed in Cl^−^ of all HHP-treated groups and Ca^2+^ of 400 and 600 MPa-treated groups (*p* < 0.05). The Ca^2+^ and Cl^−^ contents in all crayfish groups significantly increased after storing for 15 d at 4 °C, while significant decreases in PO_4_^3−^ contents were observed in all crayfish groups (*p* < 0.05). There were almost no significant changes in inorganic ion contents of crayfish groups stored at −20 °C, except for Zn^2+^ in 200 MPa treatment. HHP could cause changes in the biological membrane permeability of crayfish cells, resulting in the permeation of extracellular substances into tissues and leakage of some metal ions [42,43]. The changes of inorganic ions caused by HHP have been reported in many aquatic foods, such as the Suminoe oyster and squid [9,21].

In living organisms, inorganic ions play important roles in regulating cell osmotic pressure, acting as an important component of functional proteins, and so on. In food products, inorganic ions are essential auxiliary flavor components in aquatic food products. Liu et al. [14] found that the TAVs of K^+^, Na^+^, PO_4_^3−^, and Cl^−^ were >1 and were the main inorganic ions in oysters. In our study, the TAVs of PO_4_^3−^, Cl^−^, and K^+^ were more than 1 in control crayfish, while the contents of Ca^2+^ increased and reached more than its taste threshold (TAV > 1) after HHP treatment. PO_4_^3−^ could enhance the intensities of umami and sour tastes and suppress bitterness [39]. K^+^ could contribute to bitter and salty tastes [14]. Cl^−^ could increase umami and sweet tastes and decrease sour taste [14]. Ca^2+^ showed a negative correlation with EUC [9]. In our study, the decrease of PO_4_^3−^ after HHP treatment and 4 °C storage and increase of Ca^2+^ after HHP treatment suppressed the positive flavor of crayfish, while the increase of Cl^−^ after HHP treatment and 4 °C storage enhanced the positive taste.

## 3. Materials and Methods

### 3.1. Sampling, Packaging, and Pressure Treatment

More than 300 adult red claw crayfish (*C. quadricarinatus*) (~100 mm of total length; ~40 g of wet weight) were purchased from a commercial crayfish farm in Chengmai, Hainan Province, China, in June 2021. A total of 180 crayfish were selected and cleaned.

Before HHP treatments, all crayfish were individually packaged in double poly bags and then sealed with a vacuum. The 180 packed crayfish were randomly divided into control and three HHP-treated (including 200 MPa, 400 MPa, and 600 MPa) groups (45 crayfish for each group), respectively. HHP was performed with HPP.L1-66/5 type ultra-high-pressure equipment with a 5-L cylindrical pressure vessel (Tianjin Huatai-Senmiao Bioengineering Technology Co. Ltd., Tianjin, China). HHP-treated groups were processed for 3 min at 20 °C. The pressure rates were 300 MPa/min and released within 3 s for all HHP treated groups. Crayfish of each group were then randomly divided into three sub-groups (15 crayfish for each sub-group), in which two groups were stored at 4 °C and analyzed after 0 d and 15 d. One group was stored at −20 °C and analyzed after 15 d. For each sub-group, the meat of 5 crayfish was mixed, and stored at −80 °C for analyses.

### 3.2. Fatty Acid Analysis

The fatty acids of crayfish meat were extracted with chloroform–methanol (2:1, *v*/*v*) according to the method of Folch et al. [44]. In detail, the total fatty acids were extracted with chloroform–methanol (2:1, *v*/*v*), then saponified using 0.4 M potassium hydroxide (KOH) in methanol, followed by esterification with 25% boron trifluoride ether solution in methanol, and finally the extraction of fatty acid methyl esters (FAMEs) in hexane. Then, the FAMEs were analyzed by gas chromatograph (Hewlett-Packard model HP 5890, Palo Alto, CA, USA). Identification of fatty acids was made after a comparison of their retention times with standards (Supelco 37 Component FAME Mix, Bellefonte, PA, USA).

### 3.3. Free Amino Acid Assay

FAAs in crayfish samples were determined using HPLC (Waters 2996, Waters Corporation, Milford, MA, USA) according to the method described in Liu et al. [9]. A sample of 2.5 g was homogenized in three volumes of 10% trichloroacetic acid (TCA) and centrifuged at 10,000× *g* for 15 min at 4 °C. Supernatants were then analyzed for FAAs by high-performance liquid chromatography (HPLC) in a Waters 2996 (Waters Corporation, Milford, MA, USA). The identity and quantity of each amino acid were assessed by comparing the retention times and peak areas of each amino acid standard (Sigma-Aldrich, St. Louis, MO, USA).

### 3.4. 5′-Nucleotide Assay

The 5′-nucleotides in crayfish were extracted and analyzed according to the method described by Liu et al. [9]. HPLC conditions for 5′-nucleotide analyses were as follows: injection volumes: 20 μL, mobile phase: methanol and 0.05% phosphoric acid; flow rate: 1.0 mL/min; column temperature: 30 °C; detector wavelength: 260 nm.

### 3.5. Equivalent Umami Concentration (EUC)

The EUC [g monosodium glutamate (MSG) per 100 g tissue weight] is the concentration of MSG equivalent to the umami intensity given by the mixture of MSG-like amino acids and the 5′-nucleotides, and is represented by the following equation:*Y* = Σ*a*_i_*b*_i_ + 1218(Σ*a*_i_*b*_i_) (Σ*a*_j_*b*_j_)
where *Y* equals g MSG per 100 g; *a*_i_ is the concentration (g/100 g) of each umami amino acid (Asp or Glu); *a*_j_ is the concentration (g/100 g) of each umami 5′-nucleotide (IMP, GMP and AMP); *b*_i_ is the relative umami concentration for each umami MSG to MSG (Glu, 1 and Asp, 0.077); *b*_j_ is the relative umami concentration for each umami 5′-nucleotide to IMP (IMP, 1; GMP, 2.3 and AMP, 0.18); 1218 is a synergistic constant based on the relative umami concentration of g/100 g used.

### 3.6. Organic Acid and Betaine Assay

Crayfish meat (2 g) was homogenized in 10 mL of purified water for 5 min. Then, the sample was centrifuged at 10,000× *g* for 20 min. Malic acid, lactic acid, citric acid, and succinic acid were analyzed according to the previously described method [9]. The HPLC conditions were the same as mentioned in Section 2.4 except for the detector wavelength (215 nm).

Betaine in crayfish was extracted and analyzed with the method described by Liu et al. [9]. HPLC conditions were as follows: injection volumes: 20 μL; mobile phase: 83% aqueous acetonitrile; flow rate: 0.7 mL/min; column temperature: 20 °C; detector wavelength: 260 nm.

### 3.7. Inorganic Ion Assay

The concentrations of mineral composition (Ca^2+^, Na^+^, K^+^, Mg^2+^, and Zn^2+^) in crayfish were measured using flame atomic absorption spectrophotometry (AA-6800, Shimadzu Corporation, Tokyo, Japan) following a previously described method [45]. The concentrations of PO_4_^3−^ and Cl^−^ were analyzed using an 882-ion chromatograph system according to the method described by Liu et al. [9].

### 3.8. Statistical Analysis

Data were presented as mean ± standard deviation (SD). Data of fatty acid profiles and flavor contents of all treatments were analyzed by two-way analysis of variance (ANOVA), and the means were subsequently separated by Tukey’s test. Prior to ANOVA, homogeneity of variances was tested using Levene’s test. Statistical treatment of the data was performed using the Data Processing System (DPS) statistical software.

## 4. Conclusions

This study studied the changes in fatty acids and flavor compounds of red claw crayfish treated with different HHPs and storage temperatures. The level of PUFA in crayfish decreased with the increase of pressure at 0 d, and cold storage (4 °C and 20 °C) for 15 d almost did not cause significant changes in fatty acid profiles. HHP significantly increased the FAA contents in crayfish. There were almost no significant differences in the FAA contents of crayfish stored at 4 °C and −20 °C for 15 d compared to the crayfish treated with the same HHP levels. GMP, IMP, and AMP were the main 5′-nucleotide in crayfish, and their contents decreased after HHP treatment, except for IMP in 200 MPa treatment. The contents of citric and lactic acids decreased after HHP pressure compared to control crayfish, while there was a slight decrease in succinic acid. After storing at 4 °C and −20 °C for 15 d, the citric acid contents in all groups significantly decreased, and the lactic acid contents in control and 200 MPa treatments significantly increased (*p* < 0.05). The PO_4_^3−^ contents in HHP-treated crayfish were significantly lower than those in the untreated group (*p* < 0.05), while opposite results were observed in Cl^−^ of all HHP-treated groups and Ca^2+^ of 400 and 600 MPa-treated groups (*p* < 0.05). The Ca^2+^ and Cl^−^ contents in all crayfish groups significantly increased after storage at 4 °C (*p* < 0.05), and significant decreases in PO_4_^3−^ contents were observed in all crayfish groups (*p* < 0.05). However, almost no significant changes in inorganic ion contents were found when stored at −20 °C. In conclusion, HHP at 200 MPa or 400 MPa provides minimal changes considering the contents of fatty acids and non-volatile taste active compounds in crayfish, and storage at −20 °C is more protective compared to at 4 °C.

## Figures and Tables

**Figure 1 molecules-27-05098-f001:**
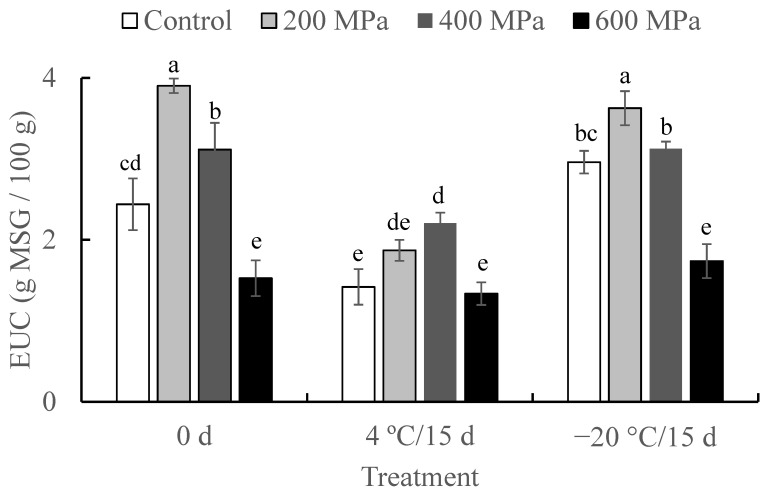
The equivalent umami concentration (EUC, g MSG/100 g wet weight) of red claw crayfish with different treatments. Bars with different letters denote significantly different (*p* < 0.05).

**Table 1 molecules-27-05098-t001:** Changes in fatty acid profiles of HHP treated red claw crayfish with different treatments.

Fatty Acid (%)	0 d				4 °C /15 d				−20 °C /15 d			
	Control	200 MPa	400 MPa	600 MP	Control	200 MPa	400 MPa	600 MP	Control	200 MPa	400 MPa	600 MP
C12:0	0.36 ± 0.02a	0.28 ± 0.03ab	0.21 ± 0.05b	0.33 ± 0.07a	0.23 ± 0.04b	0.32 ± 0.05a	0.25 ± 0.05b	0.19 ± 0.07b	0.27 ± 0.02ab	0.20 ± 0.04	0.24 ± 0.02b	0.20 ± 0.03b
C14:0	0.43 ± 0.04a	0.51 ± 0.07a	0.48 ± 0.05a	0.58 ± 0.11a	0.49 ± 0.09a	0.56 ± 0.08a	0.51 ± 0.04a	0.46 ± 0.08a	0.45 ± 0.02a	0.47 ± 0.05a	0.51 ± 0.06a	0.49 ± 0.04a
C15:0	0.98 ± 0.04a	1.01 ± 0.06a	0.96 ± 0.13a	1.23 ± 0.09a	1.13 ± 0.14a	1.18 ± 0.07a	0.97 ± 0.09a	0.97 ± 0.13a	1.10 ± 0.07a	1.02 ± 0.05a	1.12 ± 0.11a	1.07 ± 0.09a
C16:0	6.30 ± 0.34c	7.04 ± 0.45bc	7.25 ± 0.19bc	8.01 ± 0.45a	8.02 ± 0.25a	7.88 ± 0.61ab	8.22 ± 0.11a	7.97 ± 0.29ab	7.43 ± 0.12b	7.52 ± 0.22b	8.04 ± 0.18a	8.08 ± 0.35a
C17:0	1.66 ± 0.05a	1.54 ± 0.11a	1.35 ± 0.06a	1.65 ± 0.07a	1.44 ± 0.05a	1.70 ± 0.08a	1.53 ± 0.06a	1.36 ± 0.06a	1.58 ± 0.06a	1.31 ± 0.12a	1.36 ± 0.09a	1.51 ± 0.21a
C18:0	6.26 ± 0.71a	6.05 ± 0.09a	6.24 ± 0.22a	6.54 ± 0.31a	6.68 ± 0.18a	6.08 ± 0.25a	6.04 ± 0.31a	6.36 ± 0.11a	5.93 ± 0.26a	6.51 ± 0.41a	6.05 ± 0.18a	5.96 ± 0.25a
C20:0	1.09 ± 0.04a	0.88 ± 0.03ab	0.65 ± 0.07b	1.02 ± 0.12a	0.75 ± 0.09ab	0.94 ± 0.05ab	0.80 ± 0.07ab	0.65 ± 0.07b	0.86 ± 0.05ab	0.68 ± 0.04b	0.79 ± 0.08ab	0.66 ± 0.06b
ΣSFA	17.08 ± 1.03b	17.30 ± 0.78b	17.14 ± 0.66b	19.36 ± 0.98a	18.74 ± 0.74ab	18.67 ± 1.15ab	18.32 ± 0.66ab	17.96 ± 0.80b	17.64 ± 0.47b	17.71 ± 0.93b	18.11 ± 0.68ab	17.96 ± 0.93ab
C16:1n-7	2.98 ± 0.11a	2.78 ± 0.11a	2.60 ± 0.09a	3.14 ± 0.21a	2.87 ± 0.17a	3.17 ± 0.22a	3.25 ± 0.28a	2.86 ± 0.11a	3.09 ± 0.09a	2.71 ± 0.11a	2.98 ± 0.21a	2.91 ± 0.25a
C16:1n-5	0.65 ± 0.08a	0.51 ± 0.03a	0.39 ± 0.06a	0.59 ± 0.04a	0.44 ± 0.06a	0.58 ± 0.02a	0.49 ± 0.04a	0.40 ± 0.02a	0.53 ± 0.05a	0.39 ± 0.03a	0.48 ± 0.06a	0.42 ± 0.02a
C17:1n-7	1.27 ± 0.02a	1.10 ± 0.03a	0.93 ± 0.06a	1.28 ± 0.07a	1.07 ± 0.05a	1.23 ± 0.07a	1.13 ± 0.09a	1.04 ± 0.07a	1.16 ± 0.07a	0.95 ± 0.10a	1.06 ± 0.06a	1.03 ± 0.08a
C18:1n-9	16.59 ± 1.12b	17.11 ± 0.56b	18.26 ± 0.66ab	18.56 ± 1.23ab	19.07 ± 0.83ab	18.31 ± 0.79ab	18.93 ± 1.37ab	20.64 ± 0.98a	17.50 ± 1.02ab	18.00 ± 0.96ab	18.44 ± 1.17ab	19.62 ± 0.86a
ΣMUFA	21.49 ± 1.10c	21.50 ± 0.64c	22.19 ± 0.75bc	23.57 ± 0.98ab	23.44 ± 0.86ab	23.29 ± 0.84ab	23.79 ± 1.30ab	24.94 ± 1.04a	22.26 ± 1.03bc	22.06 ± 0.98bc	22.97 ± 1.26bc	23.98 ± 0.95ab
C18:2n-6	13.18 ± 0.78c	15.18 ± 0.31ab	15.70 ± 1.13ab	15.10 ± 0.95ab	15.86 ± 0.79ab	15.04 ± 1.01abc	15.83 ± 0.79ab	15.80 ± 0.63ab	14.89 ± 1.04abc	16.63 ± 0.77a	16.71 ± 0.83a	16.56 ± 0.49a
C18:3n-6	1.36 ± 0.04a	1.05 ± 0.03ab	0.79 ± 0.05c	1.22 ± 0.06a	0.93 ± 0.07ab	1.20 ± 0.04a	1.00 ± 0.11ab	0.81 ± 0.06c	1.09 ± 0.07ab	0.82 ± 0.04	0.95 ± 0.07ab	0.83 ± 0.05ab
C18:3n-5	1.81 ± 0.05a	1.36 ± 0.05ab	0.94 ± 0.08c	1.53 ± 0.14ab	1.12 ± 0.05c	1.53 ± 0.06ab	1.17 ± 0.09c	0.92 ± 0.08c	1.33 ± 0.08ab	1.12 ± 0.07c	1.24 ± 0.13c	0.96 ± 0.08c
C18:3n-3	2.32 ± 0.10a	2.11 ± 0.22a	2.73 ± 0.08a	2.01 ± 0.25a	2.01 ± 0.18a	2.43 ± 0.16a	2.20 ± 0.26a	1.76 ± 0.14a	2.21 ± 0.14a	1.81 ± 0.12a	2.07 ± 0.09a	2.15 ± 0.21a
C18:2n-7	1.60 ± 0.06a	1.33 ± 0.05a	1.01 ± 0.08a	1.70 ± 0.07a	1.25 ± 0.09a	1.59 ± 0.05a	1.20 ± 0.07a	1.05 ± 0.08a	1.32 ± 0.09a	1.08 ± 0.05a	1.24 ± 0.11a	1.00 ± 0.07a
C20:2n-6	2.76 ± 0.07a	2.88 ± 0.06a	2.77 ± 0.11a	3.05 ± 0.14a	2.78 ± 0.06a	3.03 ± 0.21a	2.82 ± 0.07a	2.86 ± 0.21a	2.85 ± 0.08a	2.88 ± 0.09a	3.01 ± 0.26a	2.92 ± 0.17a
C20:4n-6 (ARA)	10.35 ± 0.33a	9.99 ± 0.23a	8.06 ± 0.36bc	7.60 ± 0.27c	8.00 ± 0.18bc	8.92 ± 0.31ab	7.65 ± 0.15c	8.25 ± 0.22bc	8.65 ± 0.37bc	7.92 ± 0.29c	7.47 ± 0.26c	8.11 ± 0.19bc
C20:5n-3 (EPA)	22.23 ± 1.09a	21.22 ± 1.10ab	22.19 ± 1.16a	19.06 ± 0.78bc	20.62 ± 1.22ab	18.55 ± 0.60c	20.43 ± 0.75ab	19.73 ± 1.01bc	22.40 ± 0.85a	22.09 ± 0.77a	20.59 ± 0.82ab	20.10 ± 0.46abc
C22:6n-3 (DHA)	5.81 ± 0.22a	6.07 ± 0.18a	6.37 ± 0.32a	5.80 ± 0.27a	5.25 ± 0.18a	5.75 ± 0.36a	5.59 ± 0.16a	5.93 ± 0.31a	5.36 ± 0.41a	5.89 ± 0.24a	5.63 ± 0.19a	5.44 ± 0.26a
ΣPUFA	61.44 ± 2.26a	61.19 ± 2.10a	60.68 ± 2.57ab	57.07 ± 2.18b	57.82 ± 1.68b	58.04 ± 1.96b	57.90 ± 2.14b	57.50 ± 2.41b	60.10 ± 2.51ab	60.23 ± 2.50ab	58.92 ± 2.17b	58.07 ± 1.76b
Σn-3	30.37 ± 1.23ab	29.40 ± 1.32ab	31.39 ± 1.44a	27.07 ± 1.16c	27.88 ± 1.37bc	26.73 ± 1.04c	28.22 ± 1.13bc	27.42 ± 1.34bc	29.97 ± 1.32ab	29.79 ± 1.08ab	28.30 ± 1.01bc	27.69 ± 0.89bc
Σn-6	27.65 ± 1.19ab	29.10 ± 0.57a	27.33 ± 1.36ab	26.77 ± 1.12b	27.57 ± 0.97ab	28.20 ± 1.36ab	27.30 ± 1.02ab	27.71 ± 1.12ab	27.47 ± 1.23ab	28.24 ± 1.13ab	28.15 ± 1.23ab	28.41 ± 0.75ab
Σn-3/Σn-6	1.10	1.01	1.15	1.01	1.01	0.95	1.03	0.99	1.09	1.05	1.01	0.97

Data are presented as mean ± standard deviation. Different letters within the same row denote significant differences (*p*
*<* 0.05). SFA—saturated fatty acids; MUFA—monounsaturated fatty acids; PUFA—polyunsaturated fatty acids. Σn-3: 8:3n3; 20:4n-3; 20:5n-3; 22:6n-3. Σn-6: 18:2n6; 18:3n-6; 20:2n6; 20:4n-6.

**Table 2 molecules-27-05098-t002:** The contents and taste attributes of FAAs in red claw crayfish with different treatments.

FAA (mg/g)	0 d				4 °C/15 d				−20 °C/15 d				Taste Attribute
	Control	200 MPa	400 MPa	600 MPa	Control	200 MPa	400 MPa	600 MPa	Control	200 MPa	400 MPa	600 MPa	
Aspartic acid	0.02 ± 0.00c	0.03 ± 0.01c	0.06 ± 0.01ab	0.08 ± 0.02a	0.03 ± 0.01c	0.03 ± 0.01c	0.10 ± 0.01a	0.09 ± 0.02a	0.02 ± 0.02c	0.02 ± 0.01c	0.04 ± 0.01bc	0.09 ± 0.01a	Umami (+)
Glutamic acid	0.10 ± 0.01e	0.23 ± 0.02d	0.42 ± 0.02bc	0.28 ± 0.01cd	0.13 ± 0.02e	0.21 ± 0.03d	0.58 ± 0.03a	0.47 ± 0.02ab	0.11 ± 0.02e	0.23 ± 0.03d	0.34 ± 0.03c	0.27 ± 0.02cd	Umami (+)
ΣUAA	0.12 ± 0.01f	0.26 ± 0.03de	0.48 ± 0.03bc	0.36 ± 0.02cd	0.16 ± 0..03ef	0.24 ± 0.03de	0.68 ± 0.04a	0.56 ± 0.03ab	0.13 ± 0.03f	0.25 ± 0.04de	0.38 ± 0.04cd	0.36 ± 0.03cd	
Threonine	0.12 ± 0.02b	0.30 ± 0.03a	0.27 ± 0.04a	0.41 ± 0.02a	0.26 ± 0.03a	0.30 ± 0.04a	0.18 ± 0.02ab	0.30 ± 0.03a	0.23 ± 0.03ab	0.20 ± 0.02ab	0.31 ± 0.05a	0.41 ± 0.03a	Sweet (+)
Serine	0.24 ± 0.02a	0.29 ± 0.02a	0.17 ± 0.02a	0.26 ± 0.03a	0.25 ± 0.01a	0.19 ± 0.02a	0.08 ± 0.03b	0.08 ± 0.03b	0.25 ± 0.02a	0.23 ± 0.03a	0.25 ± 0.01a	0.23 ± 0.04a	Sweet (+)
Glycine	2.22 ± 0.11a	1.07 ± 0.08b	1.27 ± 0.06b	1.03 ± 0.09b	0.98 ± 0.08b	1.56 ± 0.13ab	1.47 ± 0.06ab	1.12 ± 0.07b	0.96 ± 0.08b	1.23 ± 0.05b	1.57 ± 0.06ab	1.23 ± 0.04b	Sweet (+)
Alanine	0.99 ± 0.12b	1.73 ± 0.05ab	1.54 ± 0.05ab	1.47 ± 0.03ab	1.60 ± 0.08ab	1.99 ± 0.06a	2.00 ± 0.22a	1.77 ± 0.24ab	1.09 ± 0.09b	1.20 ± 0.04b	1.64 ± 0.22ab	1.64 ± 0.06ab	Sweet (+)
Proline	0.23 ± 0.03c	0.48 ± 0.04bc	0.54 ± 0.03ab	0.37 ± 0.05bc	0.55 ± 0.03ab	0.57 ± 0.05ab	0.86 ± 0.04a	0.55 ± 0.05ab	0.41 ± 0.05bc	0.42 ± 0.04bc	0.58 ± 0.06ab	0.41 ± 0.05bc	Sweet/bitter (+)
Glutamine	0.99 ± 0.14c	1.56 ± 0.08ab	1.76 ± 0.06a	1.52 ± 0.06ab	0.84 ± 0.07bc	0.91 ± 0.08c	1.01 ± 0.05c	1.33 ± 0.09bc	1.24 ± 0.12bc	1.35 ± 0.07bc	1.87 ± 0.11a	1.85 ± 0.08a	Sweet (+)
Asparagine	0.24 ± 0.04b	0.40 ± 0.04ab	0.53 ± 0.02ab	0.61 ± 0.05a	0.24 ± 0.04b	0.54 ± 0.03ab	0.38 ± 0.04b	0.49 ± 0.05ab	0.35 ± 0.05b	0.37 ± 0.07b	0.59 ± 0.05ab	0.66 ± 0.06a	Sweet (+)
ΣSAA	5.03 ± 0.44c	5.83 ± 0.37bc	6.08 ± 0.22ab	5.67 ± 0.26bc	4.72 ± 0.34c	6.06 ± 0.30ab	5.98 ± 0.42ab	5.64 ± 0.54bc	4.53 ± 0.41c	5.00 ± 0.28c	6.81 ± 0.51a	6.52 ± 0.33ab	
Valine	0.10 ± 0.02c	0.24 ± 0.02ab	0.21 ± 0.02abc	0.23 ± 0.03abc	0.26 ± 0.03ab	0.36 ± 0.02a	0.40 ± 0.04a	0.21 ± 0.02abc	0.16 ± 0.02bc	0.17 ± 0.03bc	0.23 ± 0.04abc	0.21 ± 0.04abc	Bitter/sweet (−)
Methionine	0.11 ± 0.02b	0.23 ± 0.04a	0.20 ± 0.02ab	0.23 ± 0.01a	0.18 ± 0.03ab	0.25 ± 0.01a	0.28 ± 0.01a	0.20 ± 0.04ab	0.12 ± 0.01b	0.15 ± 0.03b	0.21 ± 0.04ab	0.24 ± 0.03a	Bitter/sweet/sulfurous (−)
Leucine	0.11 ± 0.04c	0.28 ± 0.05ab	0.21 ± 0.03bc	0.23 ± 0.05bc	0.31 ± 0.06ab	0.49 ± 0.08a	0.52 ± 0.09a	0.24 ± 0.07bc	0.15 ± 0.03c	0.18 ± 0.06bc	0.22 ± 0.04bc	0.23 ± 0.01bc	Bitter (−)
Tryptophan	0.24 ± 0.04e	1.47 ± 0.06d	2.36 ± 0.21cd	0.15 ± 0.05e	2.01 ± 0.23cd	3.15 ± 0.31b	5.94 ± 0.62a	2.84 ± 0.14c	0.49 ± 0.05e	1.32 ± 0.07d	2.86 ± 0.11c	0.22 ± 0.06e	Bitter (−)
Phenylalanine	0.16 ± 0.03bc	0.13 ± 0.02c	0.24 ± 0.04b	0.25 ± 0.02b	0.25 ± 0.04b	0.39 ± 0.05a	0.44 ± 0.06a	0.31 ± 0.04ab	0.16 ± 0.03bc	0.19 ± 0.06bc	0.23 ± 0.05b	0.25 ± 0.04b	Bitter (−)
Lysine	0.19 ± 0.06c	0.48 ± 0.03b	0.44 ± 0.07b	0.46 ± 0.07b	0.27 ± 0.05bc	0.67 ± 0.07a	0.74 ± 0.04a	0.45 ± 0.06b	0.24 ± 0.04bc	0.30 ± 0.05bc	0.45 ± 0.06b	0.58 ± 0.04ab	Bitter/sweet (−)
Argnine	6.26 ± 0.56b	5.70 ± 0.36bc	5.83 ± 0.44bc	8.47 ± 0.48a	3.00 ± 0.20d	3.09 ± 0.19d	2.24 ± 0.22e	5.16 ± 0.19c	5.44 ± 0.45bc	4.98 ± 0.26c	5.98 ± 0.41b	8.28 ± 0.59a	Bitter/sweet (−)
Isoleucine	0.06 ± 0.03b	0.16 ± 0.05ab	0.13 ± 0.01ab	0.14 ± 0.04ab	0.17 ± 0.02ab	0.26 ± 0.06a	0.29 ± 0.04a	0.14 ± 0.04ab	0.09 ± 0.06b	0.10 ± 0.04b	0.15 ± 0.05ab	0.14 ± 0.03ab	Bitter (−)
Histidine	0.34 ± 0.04b	0.49 ± 0.02a	0.32 ± 0.05b	0.37 ± 0.06ab	0.26 ± 0.03b	0.53 ± 0.04a	0.38 ± 0.03ab	0.42 ± 0.06ab	0.31 ± 0.03b	0.34 ± 0.06b	0.37 ± 0.03ab	0.43 ± 0.06ab	Bitter (−)
Tyrosine	0.15 ± 0.03a	0.24 ± 0.06a	0.23 ± 0.05a	0.26 ± 0.06a	0.24 ± 0.04a	0.32 ± 0.02a	0.31 ± 0.06a	0.21 ± 0.07a	0.16 ± 0.04a	0.17 ± 0.06a	0.15 ± 0.04a	0.33 ± 0.07a	Bitter (−)
Cysteine	0.03 ± 0.02a	0.05 ± 0.03a	0.05 ± 0.01a	ND	0.03 ± 0.01a	0.04 ± 0.02a	0.04 ± 0.03a	0.03 ± 0.01a	0.03 ± 0.02a	0.03 ± 0.02a	0.04 ± 0.01a	ND	Bitter/sweet/sulfurous (−)
ΣBAA	7.75 ± 0.87d	9.37 ± 0.70bc	10.22 ± 0.87ab	10.79 ± 0.80ab	6.98 ± 0.66d	9.55 ± 0.78bc	11.58 ± 1.01a	10.21 ± 0.54ab	7.35 ± 0.36d	7.93 ± 0.67cd	10.89 ± 0.76ab	10.91 ± 0.81ab	
ΣFAA	12.90 ± 1.21d	15.46 ± 0.94bc	16.78 ± 1.01ab	16.82 ± 0.95ab	11.86 ± 0.89d	15.85 ± 0.94bc	18.44 ± 1.23a	16.41 ± 1.04ab	12.01 ± 0.72d	13.06 ± 0.89d	18.08 ± 1.13a	17.86 ± 0.99a	

Data are presented as mean ± standard deviation. Different letters within the same row denote significant differences (*p*
*<* 0.05). ND: none detected. ΣUAA—total umami amino acid; ΣSAA—total sweet amino acid; ΣBAA—total bitter amino acid; ΣFAA—total free amino acid. +—pleasant taste; −—unpleasant taste.

**Table 3 molecules-27-05098-t003:** The concentrations of 5′-nucleotides in red claw crayfish with different treatments.

Nucleotides (mg/100 g)	0 d				4 °C/15 d				−20 °C/15 d			
	Control	200 MPa	400 MPa	600 MPa	Control	200 MPa	400 MPa	600 MPa	Control	200 MPa	400 MPa	600 MPa
CMP	0.48 ± 0.21b	0.63 ± 0.51ab	0.74 ± 0.26ab	0.68 ± 0.06ab	0.54 ± 0.16b	0.54 ± 0.17b	0.86 ± 0.11ab	0.48 ± 0.16b	0.49 ± 0.09b	0.54 ± 0.13b	0.71 ± 0.33ab	1.13 ± 0.09a
UMP	ND	0.39 ± 0.09d	0.69 ± 0.11cd	0.80 ± 0.13cd	1.02 ± 0.11bc	0.73 ± 0.09cd	1.32 ± 0.91ab	0.72 ± 0.22cd	0.33 ± 0.17d	0.45 ± 0.16d	1.25 ± 0.24ab	2.45 ± 0.91a
GMP	57.15 ± 3.48a	25.46 ± 3.77b	16.14 ± 2.97bc	16.25 ± 2.31bc	18.10 ± 3.05e	2.83 ± 1.50e	4.83 ± 1.04de	8.91 ± 1.88cd	50.02 ± 2.48a	23.24 ± 2.49b	21.43 ± 2.99b	17.05 ± 3.67bc
IMP	42.27 ± 3.39b	69.61 ± 5.08a	14.31 ± 2.79c	ND	77.48 ± 4.52a	61.78 ± 5.11a	9.75 ± 1.55c	1.04 ± 0.09d	76.11 ± 7.32a	63.40 ± 5.17a	12.82 ± 3.12c	1.06 ± 0.55d
AMP	51.80 ± 2.98ab	40.52 ± 3.61bc	46.40 ± 7.99bc	31.68 ± 5.32c	6.90 ± 3.32d	3.94 ± 1.57d	6.80 ± 2.79d	4.97 ± 3.52d	65.54 ± 5.48a	46.26 ± 4.42bc	64.56 ± 7.41a	55.15 ± 4.32ab

Data are presented as mean ± standard deviation. Different letters within the same row denote significant differences (*p*
*<* 0.05). ND—none detected.

**Table 4 molecules-27-05098-t004:** The concentrations of organic acids and betaine in red claw crayfish with different treatments.

Organic Acid/Betaine (mg/g)	b				4 °C/15 d				−20 °C/15 d			
	Control	200 MPa	400 MPa	600 MP	Control	200 MPa	400 MPa	600 MP	Control	200 MPa	400 MPa	600 MP
Malic acid	0.16 ± 0.03ef	0.09 ± 0.03f	0.07 ± 0.01f	0.12 ± 0.02ef	1.41 ± 0.09a	0.97 ± 0.06b	0.63 ± 0.06c	0.15 ± 0.02ef	0.32 ± 0.03d	0.22 ± 0.02de	0.02 ± 0.02	0.15 ± 0.09ef
Lactic acid	2.67 ± 0.12d	2.53 ± 0.21d	1.14 ± 0.05e	1.40 ± 0.08e	6.24 ± 0.35a	4.22 ± 0.18c	0.83 ± 0.08e	1.35 ± 0.09e	5.43 ± 0.36b	6.41 ± 0.28a	1.22 ± 0.12e	1.26 ± 0.09e
Citric acid	59.82 ± 2.17a	11.47 ± 1.04d	11.69 ± 0.95d	17.08 ± 4.42cd	2.78 ± 0.26e	2.61 ± 0.44e	2.61 ± 0.28e	5.88 ± 2.15e	36.75 ± 3.47b	12.24 ± 2.17d	12.74 ± 0.48d	19.78 ± 2.86c
Succinic acid	9.62 ± 0.22ab	8.74 ± 0.52bc	8.32 ± 0.38bc	7.94 ± 0.53c	9.75 ± 0.48ab	11.80 ± 0.73a	8.87 ± 0.49bc	7.60 ± 0.43c	10.47 ± 1.25ab	11.91 ± 1.39a	9.69 ± 1.04ab	7.26 ± 0.59c
Betaine	0.01 ± 0.00a	0.00 ± 0.00a	0.00 ± 0.00a	0.01 ± 0.00a	0.00 ± 0.00a	0.00 ± 0.00a	0.00 ± 0.00a	0.00 ± 0.00a	0.00 ± 0.00a	0.00 ± 0.00a	0.00 ± 0.00a	0.00 ± 0.00a

Data are presented as mean ± standard deviation. Different letters within the same row denote significant differences (*p*
*<* 0.05).

**Table 5 molecules-27-05098-t005:** The concentrations of mineral ions in red claw crayfish with different treatments.

Mineral Ions	0 d				4 °C/15 d				−20 °C/15 d			
	Control	200 MPa	400 MPa	600 MPa	Control	200 MPa	400 MPa	600 MP	Control	200 MPa	400 MPa	600 MP
Ca^2+^ (g/kg)	0.89 ± 0.07d	1.59 ± 0.36cd	1.88 ± 0.31c	1.80 ± 0.09c	3.57 ± 0.22a	3.47 ± 0.40a	2.04 ± 0.27b	2.29 ± 0.25b	0.90 ± 0.09d	1.36 ± 0.08cd	1.97 ± 0.28c	1.92 ± 0.16c
Na^+^ (g/kg)	1.35 ± 0.06b	1.87 ± 0.09ab	1.76 ± 0.26ab	1.58 ± 0.35b	2.12 ± 024a	2.01 ± 0.17a	2.07 ± 0.15a	2.24 ± 0.33a	1.53 ± 0.27b	1.56 ± 0.17b	1.73 ± 0.09ab	1.64 ± 0.08ab
K^+^ (g/kg)	3.29 ± 0.25a	2.71 ± 0.21ab	2.88 ± 0.35ab	3.07 ± 0.27a	2.43 ± 0.09bc	2.47 ± 0.14bc	2.37 ± 0.17bc	2.17 ± 0.14bc	3.07 ± 0.27a	2.86 ± 0.36ab	2.75 ± 0.09ab	2.78 ± 0.15ab
Mg^2+^ (g/kg)	0.31 ± 0.03a	0.33 ± 0.04a	0.29 ± 0.03a	0.32 ± 0.04a	0.43 ± 0.05a	0.37 ± 0.05a	0.33 ± 0.02a	0.27 ± 0.06a	0.32 ± 0.04a	0.34 ± 0.03a	0.32 ± 0.05a	0.30 ± 0.03a
Zn^2+^ (mg/kg)	34.65 ± 1.13c	39.93 ± 0.68ab	39.85 ± 1.75ab	42.01 ± 0.96a	36.75 ± 1.25bc	33.06 ± 1.06c	38.15 ± 0.84bc	42.24 ± 1.13a	32.10 ± 1.40c	36.14 ± 1.22bc	41.92 ± 0.96a	42.46 ± 1.04a
Cl^−^ (mg/g)	1.63 ± 0.05c	2.37 ± 0.06bc	2.12 ± 0.15bc	1.73 ± 0.08c	2.42 ± 0.28ab	2.53 ± 0.17ab	2.60 ± 0.09ab	3.03 ± 0.22a	1.89 ± 0.10c	1.99 ± 0.25bc	2.14 ± 0.08bc	1.90 ± 0.05c
PO_4_^3−^ (mg/g)	6.83 ± 0.09a	5.07 ± 0.25b	4.91 ± 0.12b	5.34 ± 0.31ab	0.69 ± 0.07c	0.77 ± 0.07c	1.11 ± 0.09c	0.84 ± 0.08c	6.15 ± 0.13a	5.90 ± 0.21a	5.27 ± 0.31ab	5.52 ± 0.27ab

Data are presented as mean ± standard deviation. Different letters within the same row denote significant differences (*p*
*<* 0.05).

## Data Availability

The data presented in this study are available upon request.

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
