# Peer review of "Effects of High Hydrostatic Pressure and Storage Temperature on Fatty Acids and Non-Volatile Taste Active Compounds in Red Claw Crayfish (Cherax quadricarinatus)"

_molecules, 2022, doi:10.3390/molecules27165098_

Round 1
Reviewer 1 Report
Title
The title and the aim of the study are clearly constructed.
Abstract
The abstract includes the aim of the study, methods used in the experiment and contain the principal results and conclusions.
Introduction
The introduction describes the matter of the experiment accurately and clearly states the problem being investigated.
Methods
The data is well collected. The methods are described in detail, in the way which permits the research to be replicated. The sampling is appropriate and adequately described.
Results
The results were discussed extensively, in a clear and legible way.
Discussion
They correctly interpreted and described the significance of the results for the research. Theyskillfully referred to the results of other researchers.
References
The references are accurate.
Language
The article is correctly written.
Author Response
Thanks for the recognition of this manuscript. The manuscript has been carefully revised.
Reviewer 2 Report
1. HPP has commercial value in aquatic industry applications, and Red Claw Crayfish is also very popular with consumers, so this manuscript is very important.
2. The relationship between microbial contamination and tissue structure for quality is clearly stated in the manuscript
3. Too many table fields in the manuscript are not easy to read.
4. The methyl esterification reagent is omitted in fatty acid methylation, and it is easy to produce errors only in % quantification.
5. Table3 is recommended to be presented in mg/100g
Author Response
Question 1: Too many table fields in the manuscript are not easy to read.
Respond: Because each kind of nutritional ingredient and flavor substances detected in this manuscript includes lots of items, therefore, it is more easy to read in table compared to picture. Moreover, the EUC value is used in picture.
Question 2: The methyl esterification reagent is omitted in fatty acid methylation, and it is easy to produce errors only in % quantification.
Respond: According to the reviewer’s advice, the method of fatty acid methylation were rewritten as follow: “In detail, the total fatty acids were extracted with chloroform–methanol (2:1, v/v), then saponified using 0.4 M potassium hydroxide (KOH) in methanol, followed by esterification with 25% boron trifluoride ether solution in methanol, and finally the extraction of fatty acid methyl esters (FAMEs) in hexane.”
As for the % quantification of fatty acid. Our previous study (Lin et al., 2021) has detected the lipid content (% of wet weight) of crayfish with different treatments. Therefore, in current study, we focus on the differences of fatty acid composition (% of total detected fatty acids) among different treatment. Furthermore the detection of fatty acid profile has also been widely used other articles, such as Chinese mitten crabs in Kong et al. (2012), oyster in Liu et al. (2022), etc.
- Lin, X.; Liu, C.; Cai, L.; Yang, J.; Zhou, J.; Jiang, H.; Shi, Y.; Gu, Z. Effect of high hydrostatic pressure processing on biochemical characteristics, bacterial counts, and color of the red claw crayfish Cherax quadricarinatus. Shell. Res. 2021, 40, 177–184.
- Liu, C.; Gu, Z.; Lin, X.; Wang, Y.; Wang, A.; Sun, Y.; Shi, Y. Effects of high hydrostatic pressure (HHP) and storage temperature on bacterial counts, color change, fatty acids and non-volatile taste active compounds of oysters (Crassostrea ariakensis). Food Chem. 2022, 372, 1311247.
- Kong, L; Cai, C.; Ye, Y.; Chen, D.; Wu, P.; Li, E.; Chen, L.; Song, L. Comparison of non-volatile compounds and sensory characteristics of Chinese mitten crabs (Eriocheir sinensis) reared in lakes and ponds: Potential environmental factors. Aquaculture 2012, 364-365, 96–102.
Question 3: Table3 is recommended to be presented in mg/100g
Respond: According to the reviewer’s advice, the data in table 3 has been modified.
Reviewer 3 Report
The paper entitled Effects of high hydrostatic pressure and storage temperature on Fatty acids and non-volatile taste active compounds in Red Claw Crayfish shed light on the changes that occur during HPP treatment of crayfish and during subsequent refrigerated or frozen storage of the pressure treated samples.
While analytics part is very well defined in the paper, explanations on the mechanisms that occur during HPP and subsequent refrigerated / frozen storage is less well explained. Please provide in the discussion part explanations on why these changes are taking place and why increasing/decreasing of certain compounds are noticed, especially for FFA.
I would also suggest to change title since I do not understand, for example, how malic and lactic acids could fit in the “non-volatile taste active compounds” .
Line 29: “ because it is considered a delicacy” instead of “because of its delicacy”
Line 41> replace “….and some chemicals” with “chemical preservatives”.
Line 42 HPP is not sterilization (!!!) because it has a limited effect on sporulated microorganisms, but it can be considered cold pasteurization. Please rephrase.
Table 1. Please arrange the data to be more readable (put Stdev on the same row or all of them in separate rows) because it is very difficult to read in its current form.
Lines C14:0, C15:0, C17:0 and C18:0 do not have significant differences, then C16:1n-7, C16:1n-5, C17:1n-7; C18:3n-3; C18:2n-7; C20:2n-6; C22:6n-3.
Please indicate in legend of tables and lines 324-326 the post-hoc statistical test that was applied to assess significant differences between groups.
Page 4, Lines 93 and 94 “Furthermore, the n-3 and n-6 PUFA treated with 600 MPa showed a significant decrease.” Looking at the values presented in Table 1 the statement is correct for n-3 but that are arguably significant differences between control (a,b) and 600 MPa (b). Differences, although indicated as significant, are very small, only minor, and it should be at least underlined in the text and analyzed considering also the differences between the samples and also the precision of the method.
Tabel 2. Significant differences should be added for all aminoacids ( i.e. Tyrosine, cysteine)
Please check the significant differences between tryptophan at 200 and 400 MPa. There must be an error in data. Correct typo. Make the table readable.
Line 144. Aminoacids should be written in the same way.
Table 4. Why betaine is included in the table showing organic acids? No significant differences are indicated for this component!
Page 10, line 249: “Inorganic ions are essential auxiliary flavor components in aquatic food.” Please do not limit the inorganic ions role to their contribution to flavor. What about their relationships with proteins? Even it is not the main concern of the paper it should be mentioned.
Page 11, Lines 323-327. The two-way analysis of variance do not provide significant differences. The authors should clearly indicate the post-hoc tests applied and if the preliminary conditions in analyzing variance were checked.
Conclusions lines 329-347, page 13-14. Conclusion part is not well written. The authors should try to provide more general conclusions ( i.e which treatment provide the minimal changes considering multiple criteria) and which storage is more protective.
Author Response
Question 1: While analytics part is very well defined in the paper, explanations on the mechanisms that occur during HPP and subsequent refrigerated / frozen storage is less well explained. Please provide in the discussion part explanations on why these changes are taking place and why increasing/decreasing of certain compounds are noticed, especially for FFA.
Respond: Thanks. The reason of increasing/decreasing of FFA has been added in Line 158-164. “Generally, FAAs in organism are produced from both proteolysis and certain amino acid metabolic pathways, in which proteolysis increases of FAA contents, while amino acid metabolism decreases concentration of certain amino acids. HHP treatment and refrigerated/ frozen storages could modify this progress. Therefore, these discrepant contents of FFAs could be explained by the differences in food system, storage temperature, pressure and substrates supplied by HHP-promoted proteolysis and metabolic rates.”
Question 2: I would also suggest to change title since I do not understand, for example, how malic and lactic acids could fit in the “non-volatile taste active compounds” .
Respond: In aquatic food, the organic acids are considered as one kind of important nonvolatile taste active compounds, which could provide positive acid taste. Therefore organic acids, including both malic and lactic acids, are usually detected in evaluating the taste-active compositions of aquatic food. For example, the lactic acid content was detected in squid muscles (Yue et al., 2016) and swimming crab Portunus trituberculatus (Chen et al., 2022); the malic and lactic acid contents were detected in Chinese shrimp Fenneropenaeus chinensis (Chen et al., 2021), and oyster (Liu et al., 2022) .
- Yue, J.; Zhang, Y.; Jin, Y.; Deng, Y.; Zhao, Y. Impact of high hydrostatic pressure on non-volatile and volatile compounds of squid muscles. Food Chem. 2016, 194, 12–19.
- Chen, W.; Li, X.; Zhao, Y.; Chen, S.; Yao, H.; Wang, H.; Wang, C.; Wu, Q. Effects of short-term low salinity stress on non-volatile flavor substances of muscle and hepatopancreas in Portunus trituberculatus. Food Compos. Anal. (2022, 109, 104520.
- Liu, C.; Gu, Z.; Lin, X.; Wang, Y.; Wang, A.; Sun, Y.; Shi, Y. Effects of high hydrostatic pressure (HHP) and storage temperature on bacterial counts, color change, fatty acids and non-volatile taste active compounds of oysters (Crassostrea ariakensis). Food Chem. 2022, 372, 1311247.
- Chen, L.; Zeng, W.; Rong, Y.; Lou, B. Characterisation of taste-active compositions, umami attributes and aroma compounds in Chinese shrimp. J. Food Sci. Tech. 2021, 56, 6311–6321.
Question 3: Line 29: “ because it is considered a delicacy” instead of “because of its delicacy”
Respond: This sentence has been changed.
Question 4: Line 41> replace “….and some chemicals” with “chemical preservatives”.
Respond: Thanks, the mistake has been corrected.
Question 5: Line 42 HPP is not sterilization (!!!) because it has a limited effect on sporulated microorganisms, but it can be considered cold pasteurization. Please rephrase.
Respond: Thanks. This sentence has been rewritten as “HHP is a non-thermal technology that has a potential for food preservation and shelf-life extension, while maintaining the natural characteristics of food”.
Question 6: Table 1. Please arrange the data to be more readable (put Stdev on the same row or all of them in separate rows) because it is very difficult to read in its current form.
Respond: I am sorry. The format of table 1 has been adjusted.
Question 7: Lines C14:0, C15:0, C17:0 and C18:0 do not have significant differences, then C16:1n-7, C16:1n-5, C17:1n-7; C18:3n-3; C18:2n-7; C20:2n-6; C22:6n-3.
Respond: In previous manuscript, there were no letters in data without significant difference within the same row. According to reviewer’s advice, the letters (significance) has also been added in the above rows.
Question 8: Please indicate in legend of tables and lines 324-326 the post-hoc statistical test that was applied to assess significant differences between groups.
Respond: Thanks. The statistical analysis was shown as “Data were presented as mean ± standard deviation (SD). Data of fatty acid profiles and flavor contents of all treatments were analyzed by two-way analysis of variance (ANOVA), and means were subsequently separated by Tukey's test. Prior to ANOVA, homogeneity of variances was tested using the Levene's Test. Statistical treatment of the data was performed using the Data Processing System (DPS) statistical software.”
Question 9: Page 4, Lines 93 and 94 “Furthermore, the n-3 and n-6 PUFA treated with 600 MPa showed a significant decrease.” Looking at the values presented in Table 1 the statement is correct for n-3 but that are arguably significant differences between control (a,b) and 600 MPa (b). Differences, although indicated as significant, are very small, only minor, and it should be at least underlined in the text and analyzed considering also the differences between the samples and also the precision of the method.
Respond: Thanks. The sentence has been rewritten as “the n-3 PUFAs (mainly contributed by EPA) in crayfish treated with 600 MPa showed a significant decrease compared to the untreated group (P < 0.05).” As the possibility mentioned by reviewer, differences between the samples and the precision of the method might be reason caused the decrease of n-3 PUFA in 600 MPa treated crayfish. However, after 15-d storage at 4/-20°C, the n-3 PUFAs in crayfish treated with 600 MPa also showed slightly decrease compared with control group, though no significant differences were observed. Furthermore, the changes in the fatty acid profiles has also been reported in other aquatic food after HHP treatment. Therefore, we concluded that this result was caused by higher pressure (600 MPa).
Question 10: Tabel 2. Significant differences should be added for all aminoacids ( i.e. Tyrosine, cysteine)
Respond: As shown in question 7, the significant differences have been added.
Question 11: Please check the significant differences between tryptophan at 200 and 400 MPa. There must be an error in data. Correct typo. Make the table readable.
Respond: Thanks. After carefully checking the original data (showed in attechment), the tryptophan at 200 and 400 MPa showed significantly higher than control and that at 600 MPa. The original data of FAAs of control, 200, 400 and 600 MPa at 0 d are shown as follow. Furthermore, the format of table 2 has been adjusted.
Question 12: Line 144. Amino acids should be written in the same way.
Respond: thanks. The mistake in Table 2 of “tyrptophan” has been corrected into “tryptophan”.
Question 13: Table 4. Why betaine is included in the table showing organic acids? No significant differences are indicated for this component!
Respond: Thanks. The presentation of betaine and organic acids has been rewritten, which are shown in “Table 4”, “2.4. Comparison of organic acids and betaine in crayfish” (As shown in Table 4, the concentrations of betaine were almost undetectable in crayfish, while …), and “3.6. Organic acid and betaine assay”. Furthermore, the significant differences of betaine have been added.
Question 14: Page 10, line 249: “Inorganic ions are essential auxiliary flavor components in aquatic food.” Please do not limit the inorganic ions role to their contribution to flavor. What about their relationships with proteins? Even it is not the main concern of the paper it should be mentioned.
Respond: According to the reviewer’s advice, the function of inorganic ions in living organisms has been added in Line 255-256. In detail, “In living organisms, inorganic ions play important roles in regulating cell osmotic pressure, acting as an important component of functional proteins, and so on.”
Question 15: Page 11, Lines 323-327. The two-way analysis of variance do not provide significant differences. The authors should clearly indicate the post-hoc tests applied and if the preliminary conditions in analyzing variance were checked.
Respond: The statistical analysis in this manuscript has been rewritten as follow, “Data were presented as mean ± standard deviation (SD). Data of fatty acid profiles and flavor contents of all treatments were analyzed by two-way analysis of variance (ANOVA), and means were subsequently separated by Tukey's test. Prior to ANOVA, homogeneity of variances was tested using the Levene's Test. Statistical treatment of the data was performed using the Data Processing System (DPS) statistical software.”
Question 16: Conclusions lines 329-347, page 13-14. Conclusion part is not well written. The authors should try to provide more general conclusions ( i.e which treatment provide the minimal changes considering multiple criteria) and which storage is more protective.
Respond: According to the reviewer’s advice, the conclusion has been rewritten as follow, “In conclusion, HHP at 200 MPa or 400 MPa provides the minimal changes considering the contents of fatty acids and non-volatile taste active compounds in crayfish, and storage at -20°C is more protective compared to at 4°C.”

Reviewer 4 Report
Dear authors,
The manuscript entitled “Effects of High Hydrostatic Pressure and Storage Temperature on Fatty Acids and Non-Volatile Taste Active Compounds in Red Claw Crayfish (Cherax Quadricarinatus)" is an interesting topic that could be of interest for readers; however, I consider that it lacks scientific solidity. It is necessary to explain from a scientific point of view the reason for their results.
Please see the attachment.

Author Response
Question 1: In the introduction the authors mentioned some microbiological aspects; however, the objective of the study has nothing to do with the microbiological aspect. Therefore, the introduction should be focused on the importance of fatty acids and non-volatile compounds. Also, you should cite any similar studies, if any. How do you justify your study?
Respond: As we all known, the most important purpose of HHP technology used in food produce is of inactivation of microorganisms and shelf-life extension. In the introduction part of our manuscript, these examples were presented in order to show effectiveness and differences in different aquatic species when treated with HHP.
As for the introduction of fatty acids and non-volatile compounds, this part has been added at the third paragraph. In detail, “Although the quality of aquatic food treated by HHP is less affected compared to conventional thermal processing, HHP, especially at higher pressure, also changes the appearance, nutrient and flavor compounds [11,17]. Recently, the effects of HHP on quality of many aquatic animals, such as such as shrimps, crabs, oysters, bay scallop, squids, and fish, has been reported [18-23]. According to these studies, the optimum HHP conditions were various among different aquatic species. Therefore, it is vital to define proper HHP treatment conditions to improve food safety and prevent the damage of nutrition and flavor components in raw aquatic products.”
Question 2: In the results and discussion section, there is no explanation of the findings reported in this study. It is necessary to explain from a scientific point of view the reason for their results.
Respond: Thanks. According to the reviewer’s advice, more information about the reason for our results has been added in some paragraph of “results and discussion section”.
For example:
In “2.1. Fatty acid compositions”, the reason why profiles of fatty acids showed different tendency has been discussed, “In this study, the higher pressure (≥ 600 MPa) could cause some changes in the fatty acid profiles. Yi et al. reported that the fat content of Pacific white shrimp also significantly decreased after HHP treatment (550 MPa/5 min), while there were only minor changes found in the compositions of fatty acid in oysters after HHP treatment. Oysters have a stronger shell and can tolerate higher pressure compared to shrimps. Crayfish showed a cooked appearance after 600 MPa treatment, while only slight changes were observed in soft tissues of oysters treated with the same pressure. The cooked phenomenon in 600 MPa-treated crayfish might cause the changing of fatty acid profiles.”
In “2.2. Changes of FAAs and 5′-nucleotide contents in crayfish”, the changes of FFA after HHP and refrigerated / frozen storage were discussed. “Generally, FAAs in organism are produced from both proteolysis and certain amino acid metabolic pathways, in which proteolysis increases of FAA contents, while amino acid metabolism decreases concentration of certain amino acids. HHP treatment and refrigerated/ frozen storages could modify this progress. Therefore, these discrepant contents of FFAs could be explained by the differences in food system, storage temperature, pressure and substrates supplied by HHP-promoted proteolysis and metabolic rates.”
Question 3: Line 78. You could change “In our study” by “In the present study”
Respond: Thanks. This sentence has been changed.
Question 4: Line 87-97. Can you propose an explanation for these findings? Why does the concentration of some fatty acids increase?
Respond: In this study, the profiles of fatty acids were detected after different HHP and storage treatments. As shown in the manuscript, cooked phenomenon in crayfish meat was observed after higher pressure treatment (600 MPa), which might cause the degradation of some fatty acids, such as EPA. And these result has been reported in many other aquatic food, such as Pacific white shrimp treated by 550 MPa/5 min, oysters treated by 600 MPa/5 min. Therefore, some kinds of fatty acids decreased while others increased.
Round 2
Reviewer 3 Report
I think the quality of the manuscript has been significantly improved. Most of the suggestions were resolved.
Minor observations:
1. In Tables 3 CMP and UMP nucleotides do not have significant differences indicated as they should.
2. In figure 1, there is an overlapping between temperatures and the measurement unit on the O x axis.
Author Response
Question 1: In Tables 3 CMP and UMP nucleotides do not have significant differences indicated as they should.
Respond: Thanks. The significant differences of CMP and UMP contents among different treatments has been added.
Question 2: In figure 1, there is an overlapping between temperatures and the measurement unit on the O x axis.
Respond: Thanks, the overlapping in figure 1 has been deleted.
Reviewer 4 Report
Dear authors,
After reading the corrections, I consider that the observations were adequately addressed. Therefore, I contemplate that the article should be accepted.
Best regards,
Author Response
Thank you very much for the valuable advice.